# Oral Health Profiling for Young and Older Adults: A Descriptive Study

**DOI:** 10.3390/ijerph18179033

**Published:** 2021-08-27

**Authors:** Jennifer Hanthorn Conquest, John Skinner, Estie Kruger, Marc Tennant

**Affiliations:** 1School of Human Sciences, University of Western Australia, Perth 6009, Australia; estie.kruger@uwa.edu.au (E.K.); marc.tennant@uwa.edu.au (M.T.); 2Poche Centre for Indigenous Health, Faculty of Medicine and Health, University of Sydney, Sydney 2006, Australia; john.skinner@sydney.edu.au

**Keywords:** oral health education, adults, dental care

## Abstract

The purpose of this study was to trial the suitability of an oral health promotion toolkit in a chair-side setting to determine: an individual’s knowledge; understanding of oral and general health behaviour and evaluate the commitment of dental practitioners to undertake an assessment of the individual’s attitude and aptitude to undertake a home care preventive plan. All participants were 18 years and over and came from low socio-economic backgrounds in rural New South Wales, Australia. The study evaluated 59 case studies regarding their knowledge of oral and general health. The study included an oral health profiling questionnaire, based on validated oral health promotion outcome measures, a full course of dental care provided by a private dental practitioner or a dental student. Out of the 59 participants, 47% of participants cleaned their teeth twice per day, 69% used fluoride toothpaste and 47% applied the toothpaste over all the bristles. The questionnaire, based on Watt et al. (2004) verified oral health prevention outcome measures was a sound approach to determine an individual’s knowledge, understanding of oral and general health behaviour. However, dental practitioners’ commitment to assessing the individual was low.

## 1. Introduction

Evidence based and best practice strategies are used to deliver effective health services, public health messages and population health strategies to improve the wellbeing of the individual and the community at large. The terms “evidence based” and “best practice” involve analysing a method/system using the best available evidence from multiple sources by (i) turning the clinical practice into a question; (ii) systematically exploring and retrieving evidence; (iii) scrutinising the evidence for its validity; (iv) weighting the evidence; (v) incorporating the evidence into a well-thought-out decision-making process; (vi) evaluating the proof of evidence process [1]. Implementation of evidence-based care that includes oral health promotion interventions need to consider a range of implementation factors to ensure clinician buy-in and sustainability [2].

Oral health practitioners are expected to be active in population health preventive education within the limitations of normal interventional practice. Practitioners are aware that they need to consider an individual’s lifestyle and behaviours when promoting health education, as these factors have a great impact on oral conditions, including dental caries, gingivitis, and periodontal disease [3]. However, social, and socio-economic determinants far outweigh the individual’s behaviour and cultural norms [4]. It is also noteworthy that the social impact of dental disease can be categorised by limitations in social functioning and social discrimination that can limit daily life activities [5]. The most significant population cohort that is internationally recognised to be at risk of poor oral health, are people from lower socio-economic backgrounds, as they usually have lower education levels, poorer diet, poorer oral hygiene and higher levels of tobacco and alcohol use [6].

Public and population preventive health care strategies are vitally important for two reasons: quality of life and the cost burden on public health services. Public health strategies are being developed for health practitioners to transfer knowledge to the individual, with the aim of raising health literacy, increasing confidence in self-reporting health behaviour, shared decision making, self-motivation and health knowledge to support behaviour change [7].

Currently there are several adult health education tools available for dental practitioners that provide generalised messaging, such as the NSW Health messages of “Eat well, drink well, clean well, play well and stay well” [8], or the Australian Dental Association’s “Water, Water Everywhere, Watch What You Eat, and Gum Everyone?” [9].

Some toolkits add guidance tips on how to support knowledge transfer aimed to improve an individual’s general and oral health by providing a practitioner checklist for different age cohorts. For example, the Public England Health 2017 oral health toolkit suggests that for adults, “Brush at least twice daily, with a fluoridated toothpaste; reduce the amount of sugary food and drinks; and use a fluoride mouth rinse daily (0.05% NaF) at a different time to brushing” [10]. The Watt et al., 2004 toolkit combines a checklist for different age cohorts for knowledge transfer with additionally provided verified questions that a practitioner can use for clinical assessment as well as study evaluation [11].

This descriptive study focused on adults (18 years of age and over) who lived in rural New South Wales (NSW), Australia, and their responses to general health behaviour questions, their habits and knowledge for improving or maintaining good oral health.

The objective of this descriptive study was to address the research questions: “Can a verified oral health promotion toolkit be used in a chair-side setting to determine an individual’s knowledge and understanding of what behaviours maintain good oral and general health” and “Does the practitioner have the commitment in assessing the level of the individual’s attitude in maintaining preventive behaviours?”

## 2. Materials and Methods

### 2.1. Study Design

The study’s oral health education component (questionnaire) was designed as a personalised home care program that aimed to improve oral hygiene practices: brushing and flossing, dietary modifications such as limiting the frequency of consumption of sweet foods and drinks and understanding of the importance of using a fluoride toothpaste, antibacterial mouthwash, and remineralising agents like calcium phosphate pastes. The flow of the questionnaire was designed to cover (i) the intermittent access to NSW public oral health services; (ii) the ability for an individual to self-assess their oral health; (iii) their knowledge on of how to maintain good oral and general health and oral health behaviours.

Participation in completing the questionnaire was voluntary and anonymous. The questionnaires were provided to the participant at their first and last appointment prior to meeting the dental practitioner/student. The completed questionnaire was part of the oral health education discussion with the dental practitioner/dental student. The 1-page questionnaire consisted of 15 questions based on the Watt et al., 2004 “Oral Health Promotion: Evaluation Toolkit” that evaluated oral health outcomes measures. These measures were weighted by importance to develop an oral health profile and those that had a high-quality score as identified in the Toolkit [11].

The Watt et al., 2004 “Oral Health Promotion Evaluation Toolkit” (Toolkit) was utilised by this study, in preference to other toolkits, because it is specifically aimed at systematically exploring and retrieving evidence to design questions to assist oral health promotion professionals to have the ability to measure outcomes. For example, “How often do you brush your teeth? (more than 3 times a day; 3 times a day; twice a day; once a day; less than once a day; I don’t brush my teeth)” [11]. This question was then validated by its use in other studies such as O’Brian, 1993 and Walker et al., 2000 [12,13].

The study design included the formal agreement of private dental practitioners/dental students’ commitment of the participation in the study. Of particular importance the completion of the section of the questionnaire where the following data would be entered: the participant’s age, sex, plaque index score (type not specified), capped-fee diagnostic pathway and attitude to oral health (self-motivated and dentally aware; dentally aware but dependent; no motivation and low awareness). The completion of this section determined the commitment of the practitioners/dental students to formally assess the individual.

### 2.2. Study Setting

This study was conducted during 1 January 2012 to 31 March 2015 within the clinical settings of a university dental clinic and a private dental clinic in Greater Southern Health region, in New South Wales Australia.

### 2.3. Participants

The study’s adult cohort was aged 18 years and over and participants were required to meet the eligibility criteria to access the NSW public dental health system, by being a holder of a current Australian Medicare and concession card [14].

The study participants were triaged using the standard NSW Priority Oral Health Program policy according to the urgency of their dental needs [15].

### 2.4. Data Source and Analysis

The data for this study were sourced from the completed questionnaires, which were collected by the administration staff and provided to the researchers for analysis. Analysis was conducted by using Version 13 Microsoft Excel.

The data for the participants’ diagnostic pathways and dental treatment provided through the Capped-Fee Model of Care (MoC) were sourced from NSW Oral Health Data Warehouse during 1 January 2012–31 March 2015 and analysed by using SAS 9.3 and Version 13 Microsoft Excel [9].

### 2.5. Study Size

It was convenient for this study to use the same cohort in the Conquest et al., 2017 study [15]. This study provided 59 full courses of care to randomly selected participants and included oral and general health education by private dental practitioners or supervised university dental students [16]. Their preventive oral health education was based on a ratified oral health promotion toolkit [11]. The participants of this study were sourced through a formal agreement that would pay the private practitioner and university by a Capped-fee MoC [16].

The Capped-fee MoC used was as described in a previous study (Conquest et al., 2017) and provided four diagnostic pathways of (i) no active caries and no pain, (ii) active caries and no pain, (iii) active caries and pain, (iv) periodontal [16]. The dental care provide in the first three pathways was “comprehensive dental examination, radiographs, removal of calculus, fluoride prevention, oral health education, tooth extraction, two surface metallic restoration such as amalgam, one and two surface anterior restorations and one and two surface posterior restoration” [16]. The periodontal pathway provided care for acute periodontal infection, root planning and subgingival curettage and prescribed medication.

### 2.6. Quantitative Variables

The quantitative variables of this study arose from the small study size (59) and so the results cannot determine trend data for the initial age group cohorts for adults as per the Capped-Fee MoC (18–24 years; 25–34 years; 35–44 years; 45–54 years; 55–64 years; 65–74 years; 75+ years) [16].

Due to the unequal participation rate between the different age groups this study’s age bands were categorised in two groups: 18–34 years (young adults) and 35–65+ years (older adults).

### 2.7. Statistical Measures

The analysis for this descriptive study did not include formal sample size calculations and use a convenience sample. The study conforms to the STROBE guidelines of cohort, case–control, and cross-sectional studies (observational) study check list [17].

The data that contained no identification of age groups and/or diagnostic pathways were not included in the analysis.

## 3. Results

### 3.1. Study Size

Of the 59 participants, the aged groups included 30 young adults, 25 older adults, and participants whose ages were not identified. The older adults who participated in the study did not live-in residential care.

### 3.2. Questionnaire Evaluation

All the Watt et al., 2004 oral health promotion toolkit assessment categories and associated questions used in the study appeared to be understood by the participants as each question received a 59 (100%) response rate [11]. In response to the question “What type of toothbrush do you use?” three participants replied saying both soft and medium, and one participant responded to both medium and hard, thus identifying the variation in habit/purchase.

### 3.3. General Health Behaviour

The general health behaviour questions (Table 1) focused on outcome measures for healthy lifestyle habits, such as how many glasses of water a day are consumed, how many sugary drinks are consumed, number of times a day the participant eats (including snacks), and lastly smoking practices.

The range of habits and knowledge that support general health (Table 1) shows that overall, 63% of participants consume 0–4 glasses of water per day. Young adults ranked the highest in this response and older adults at 64%. Young adults drank 1–2 sugary drinks a day (47%), ate 3–4 (60%) times a day, and most did not smoke (73%). However, most older adults did not drink sugary drinks (56%), ate 3–4 times a day (56%) and the majority did not smoke (72%).

### 3.4. Oral Health Habits and Awareness

The oral health habits and awareness questions (Table 2) focused on outcome measures for oral health knowledge and habits by asking questions around best practice guidelines for maintaining good dental health, such as number of times teeth are cleaned, type of toothbrush use, type of toothpaste and quantity placed on the bristles, and type and use of mouth rinse.

Young adults clean their teeth once a day (47%), using a medium toothbrush (57%) with fluoride toothpaste (67%) and 40% covered all the bristles and 40% covered half of the bristles. Older adults clean their teeth twice day (60%) using a medium toothbrush (52%) with fluoride toothpaste (84%) that covers all the bristles (64%). Young adults reported that 50% did not use mouthwashes and the other 50% rinsed with fluoride mouthwash while, older adults did not use mouthwashes (64%).

### 3.5. Participant General Assessment

The overall assessment questions (Table 3) focused on outcome measures for service use, self-assessment of their dental health, and health literacy to maintain a good oral health status. The participant’s health literacy and attitude were determined by the practitioner/dental student.

Out of the 59 participants, 47% had their last dental appointment more than 2 years ago, 32% had their last dental appointment less than 1 year ago whilst 20% had a dental appointment less than 2 years or did not specify (Table 3). Table 3 also displays how the participants rated their oral health as fair (51%), “poor” to “very poor” (32%) and other, which included not specified (17%).

There were 32 participants (60%) who responded to the question on dental problems that they were experiencing. The responses included (for those who had received dental care in less than one year): loose fillings, decay, and sensitive teeth. For those who had not received care in almost 2 years: fillings fallen out and decay. Whilst those who had not received dental care greater than 2 years: broken teeth, lost or loose fillings, sore gums, pain and need of a denture.

The practitioners/dental students’ rate of participation to assess an individual’s attitude to their oral health was the highest rate of assessment being in the age group of young adults (54%), whilst older adults were at 19%.

Further participant analysis showed that the young adults were the most self-motivated and dentally aware (47%), as well as being the most dentally aware and dependent (37%). However, the older adults had the highest number (64%) of not being assessed (Table 3).

### 3.6. Dental Treatment Provided

The Capped-fee MoC diagnostic pathways (Figure 1) provided to the participants were based on their self-assessment of their current dental conditions. Out of the 59 participants, 9 diagnostic pathways were not identified, these being 1 (3%) in young adults and 7 (28%) in older adults and 1 (25%) with no age provided.

The diagnostic pathway of “no active caries and no pain” was provided only to 6 young adults (20%). “Active caries and no pain” was provided to 12 (40%) young adults, and 5 (20%) older adults. Treatment through the “active caries and pain” diagnostic pathway was to 6 (20%) young adults, and 9 (36%) older adults. Periodontal care was given to all age groups; 7 (23%) young adults and 5 (20%) older adults.

## 4. Discussion

This study developed a series of verified oral health and general health questions, used in a chair-side setting, based on Watt et al., 2004 “Oral Health Promotion Evaluation Toolkit” (Toolkit), coupled with assessing the dental practitioner’s commitment to evaluate the individual’s attitude in maintaining a preventive homecare program [11]. The study showed that utilising the Watt et al., 2004 Toolkit provided positive results for individual assessment [11].

Additionally, the Watt et al., Toolkit provided a check list and questions for practitioners to use for the following groups: (i) parents and caregivers of pre-school children; (ii) 12-year-old children; (iii) people over 65 years of age [11]. The study demonstrated that the participating practitioners were not consistent in assessing the individual’s attitude and knowledge of oral health and general health.

The study’s process relied heavily on the dental clinical setting implementing the chairside research, which demonstrated limited success due to many factors, led by the lack of understanding of the scientific methodology [18] and a busy clinic setting. This caused inconsistency in providing the questionnaires and completion of the oral health profiling by the private practitioners/dental students. These issues contributed to a small sample of patients in this study which is a major limitation of this study.

There has been research into the effectiveness of oral health education adult questionnaires at the chairside, as well as private practitioners’ commitment to assessing an individual’s attitude towards change in behaviour. For example, Soldani et al., 2018 conducted a systematic review analysing one-on-one oral health hygiene advice in dental clinical settings. Soldani’s review shows that out of 19 studies, 15 focused on adults, and from those, 6 used questionnaires as part of their physiological profiling [18,19]. The questionnaires (like that of the current study) covered aspects such as self-assessment of their oral health, dental knowledge and behaviour including their frequency of toothbrushing and interdental cleaning and type of toothbrush used [20,21,22]. Moreover, Münster et al., 2012 used the questionnaires as a basis for tailoring the individual’s oral health educational programme on their cognitive behavioural principles and provided intermediate, and long-term goals for their dental care [21,22]. It is unfortunate that the six studies did not disclose the questions that were asked, thus not allowing comparison to this study nor did it include self-assessment of their general health. Additionally, another aspect that these six studies have in common with this study was that they also examined plaque and bleeding indices, nonetheless the references also included dento-gingival plaque [23,24].

Conversely, research has shown that a practitioner’s commitment to chairside oral health education is impacted by various factors. Masoe et al., 2015 identified the restrictions placed on the public oral health practitioner including time to allocate assessment of an individual’s attitude, or providing in-depth preventive education is restricted due to the public sector’s focus on activity-based funding, and priority given to procedures of pain relief such as exodontia and restorations [25]. Although preventive care and education is widely acknowledged as evidence-based best practice, private dental practices are slow to implement a prevention-focused approach to dental care [26,27]. There is also a view that individuals wanted ‘value for money’ and could not see this in preventive care [26]. Private practitioners believed that some individuals are not committed enough in their oral hygiene habits to benefit from prevention [25], and, this change could also risk financial viability, require additional technology and resources, time, and a multidisciplinary supportive dental team [25,26].

The next phase for developing a dental preventive care strategy would be to continue using this study’s questionnaire, but also include recording of dento-gingival plaque, and have a longer timeframe to allow some time-trend analysis of behavioural change within the different age bands, as well as monitoring the longevity of the practitioners’ commitment. It could be considered that the best clinical environment for this phase is with collaboration between private, public and university settings.

## 5. Conclusions

The study showed that use of the Watt et al., 2004 Toolkit verified oral health prevention outcome measures was a good baseline to identify individual behaviour and knowledge of their oral and general health. Additionally, the study also demonstrated that a strong administrative process is required to obtain a high level of participation for both research questions.

## Figures and Tables

**Figure 1 ijerph-18-09033-f001:**
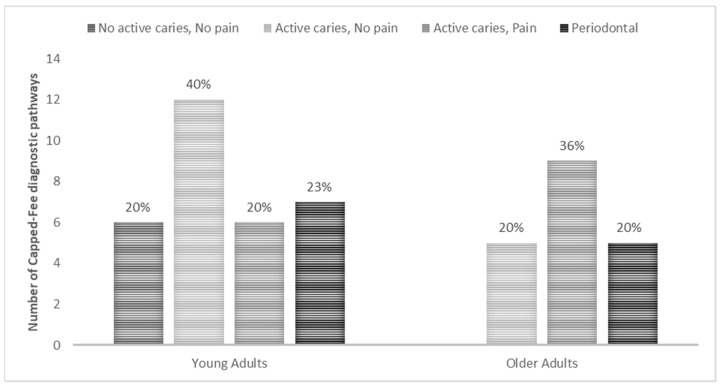
Participants’ dental care through the Capped-fee diagnostic pathways.

**Table 1 ijerph-18-09033-t001:** Participants’ habits in relation to their general health.

Cohort	Number of Glasses of Water Drunk in a Day			How Many Sugary Drinks Are Drunk in a Day			Number of Times Eating in a Day Including Snacks			Number of Times Smoking in a Day		
Age Categories	0–4 Glasses	5–7 Glasses	Other	1–2	None	Other	3–4	More than 5	Other	Zero	1–10	More than 10
Young adults	21	8	2	14	8	10	18	11	3	22	6	4
Older adults	16	6	5	9	14	3	14	5	4	18	3	4
No age provided	0	0	1	0	0	1	0	0	4	0	0	2
Total	37	14	8	23	22	14	32	16	11	40	9	10

**Table 2 ijerph-18-09033-t002:** Participants’ health in relation to their oral health.

Cohort	Number of Times Brushing Teeth in a Day			Type of Toothbrush Used			Type of Toothpaste Used			How Much Toothpaste Is Put on the Bristles of the Toothbrush			Type of Mouth Rinse Used		
Age Category	Once a Day	2 Times a Day	Other	Medium	Soft	Other	Fluoride	Whitening	Other	Covers All of	Covers Half of	Other	None	Fluoride	Other
										the Bristles					
Young adults	14	13	5	17	11	5	20	9	6	12	12	8	15	15	2
Older adults	5	15	6	13	10	5	21	4	4	16	7	2	16	3	5
No age provided	0	0	1	0	0	1	0	0	1	0	0	2	0	0	3
Total	19	28	12	30	21	11	41	13	11	28	19	12	31	18	10

**Table 3 ijerph-18-09033-t003:** Participants’ access to dental services, self-assessment of their current oral health and attitude evaluation to maintaining good oral health.

Cohort	Last Time Attended a Dental Appointment			Rating of Dental Health			Attitude to Oral Health			
Age Category	2 Years and Over	Less Than 1 Year	Other	Fair	Poor to Very Poor	Other	Self-Motivated Dentally Aware	Dentally Aware, Dependent	No Motivation, Low Awareness	Not Specified
Young adults	17	11	2	19	11	2	14	11	7	0
Older adults	11	8	6	11	8	5	5	4	1	16
No age provided	0	0	4	0	0	3	0	0	0	1
Total	28	19	12	30	19	10	19	15	8	17

## Data Availability

Our legally binding data use agreement with the custodian of the data does not allow public sharing as derived from public health application.

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
