# Peer review of "Oral Health Profiling for Young and Older Adults: A Descriptive Study"

_ijerph, 2021, doi:10.3390/ijerph18179033_

Round 1
Reviewer 1 Report
Introduction:
Can be modified to be more focused in the study related topic.
Methods:
Lines no. 113, 114 and 115 seems more into discussion than method >>
Lines 116 and 117: this sentence need to be elaborated more, as i feel it shows selection bias.
Sample size calculation:
This study lack Sample size calculation.
The authors should re calculate the Sample size according to their aim and their previous study cohort (Conquest et al., 2017). Even if this study material was derived from a previous work by the authors, each paper should include its own sample size calculation to be sure that the results found on this sample is
"For an argument" i feel that bigger sample size need to be recruited for such conclusion.
Discussion seems ok, bu i advise to menion the small sample size in this section
Author Response
Introduction:
Can be modified to be more focused in the study related topic.
Introduction has been revised in the updated manuscript.
Methods:
Lines no. 113, 114 and 115 seems more into discussion than method >>
This sentence has been deleted in the updated manuscript.
Lines 116 and 117: this sentence need to be elaborated more, as i feel it shows selection bias.
This sentence has been updated in the updated manuscript to provide this elaboration.
Sample size calculation:
This study lack Sample size calculation.
The authors should re calculate the Sample size according to their aim and their previous study cohort (Conquest et al., 2017). Even if this study material was derived from a previous work by the authors, each paper should include its own sample size calculation to be sure that the results found on this sample is
"For an argument" i feel that bigger sample size need to be recruited for such conclusion.
Sample size section has been revised in the updated manuscript.
Discussion seems ok, bu i advise to menion the small sample size in this section
Sample size issue has been included as a limitation in the discussion in the updated manuscript.
Reviewer 2 Report
Review summary Manuscript ijerph-1347207
Oral Health profiling for Young and Older Adults: a Descriptive Study
In this study the authors performed a descriptive study with a questionnaire about oral health and the procedures of dental health at 59 adults with low socio-economic backgrounds.
The aim of this study was to investigate the suitability of an oral health promotion toolkit in a chair-side setting to determine the understanding and the consequences of oral health of the participants.
Although covering an interesting topic, the present manuscript has some limitations.
- The tables are confusing and overloaded. Please make a clearer layout.
- Figure 1 is missing. Please implement it.
Reviewer comments
Title
- There is no compliant in the title.
Abstract, Introduction, Materials and methods
- There is no compliant in the abstract, introduction, materials and methods.
Results
- Page 4: 3.2. Questionnaire Evaluation
- Typing error: Please remove the 10
Line 165-168: “10 In response to the question […] “
- Page 5: 3.3. General Health Behaviour
- Confusing table: Decide to use absolute nummers or percent, not both and separate the categories with lines to get a clearer table
Line 180: Table 1. Participants’ habit in relation to their general health.
- Page 5: 3.4. Oral Health Habits and Awareness
- Confusing table: Decide to use absolute nummers or percent, not both and separate the categories with lines to get a clearer table
Line 193: Table 2. Participants’ health in relation to their oral health.
- Page 6: 3.5. Participant General Assessment
- Confusing table: Decide to use absolute nummers or percent, not both and separate the categories with lines to get a clearer table
Line 217/218: Table 3. Participants’ access to dental services, self-assessment of their current oral health and attitude evaluation to maintain good oral health.
- Page 6: 3.6. Dental Treatment Provided
- Missing Figure
Line 229: Figure 1. Participants’ dental care through the Capped-fee diagnostic pathways.
Discussion
- Interesting idea for future studies (Line 276-281).
Conclusion
- There is no complaint in the conclusion.
At present, we suggest to accept this manuscript with minor revision, when the above remarks have been addressed by the authors.
Author Response
Although covering an interesting topic, the present manuscript has some limitations.
- The tables are confusing and overloaded. Please make a clearer layout.
The Tables have been reformatted in the updated manuscript.
- Figure 1 is missing. Please implement it.
Figure 1 is included in the updated manuscript.
Reviewer comments
Title
- There is no compliant in the title.
Abstract, Introduction, Materials and methods
- There is no compliant in the abstract, introduction, materials and methods.
Results
- Page 4: 3.2. Questionnaire Evaluation
- Typing error: Please remove the 10
Line 165-168: “10 In response to the question […] “
This was a referencing formatting issue and has been addressed in the updated manuscript.
- Page 5: 3.3. General Health Behaviour
- Confusing table: Decide to use absolute nummers or percent, not both and separate the categories with lines to get a clearer table
Line 180: Table 1. Participants’ habit in relation to their general health.
The Tables have been reformatted in the updated manuscript.
- Page 5: 3.4. Oral Health Habits and Awareness
- Confusing table: Decide to use absolute nummers or percent, not both and separate the categories with lines to get a clearer table
Line 193: Table 2. Participants’ health in relation to their oral health.
The Tables have been reformatted in the updated manuscript.
- Page 6: 3.5. Participant General Assessment
- Confusing table: Decide to use absolute nummers or percent, not both and separate the categories with lines to get a clearer table
Line 217/218: Table 3. Participants’ access to dental services, self-assessment of their current oral health and attitude evaluation to maintain good oral health.
The Tables have been reformatted in the updated manuscript.
- Page 6: 3.6. Dental Treatment Provided
- Missing Figure
Line 229: Figure 1. Participants’ dental care through the Capped-fee diagnostic pathways.
Figure 1 is included in the updated manuscript.
Discussion
- Interesting idea for future studies (Line 276-281).
Conclusion
- There is no complaint in the conclusion.
At present, we suggest to accept this manuscript with minor revision, when the above remarks have been addressed by the authors.
Reviewer 3 Report
The topic may be interesting evaluating a questionnaire aimed to verify oral health prevention outcome measures and determine an individual’s knowledge, understanding of oral and general health behaviour. It is well written and the methodology clear. The only issues may be the small number of subjects included for such a type of study and the restricted area of collection, that does not allow to generalize the obtained conclusions. Moreover, it is not clear the reason why only 18-yr adults were involved.
Some more recent references on studies performing questionnaires may be added.
Author Response
The topic may be interesting evaluating a questionnaire aimed to verify oral health prevention outcome measures and determine an individual’s knowledge, understanding of oral and general health behaviour. It is well written and the methodology clear. The only issues may be the small number of subjects included for such a type of study and the restricted area of collection, that does not allow to generalize the obtained conclusions. Moreover, it is not clear the reason why only 18-yr adults were involved.
Sample size issue has been included as a limitation in the discussion in the updated manuscript.
The study included two age groups 18-34 years and 35 years and older.
Some more recent references on studies performing questionnaires may be added.
We have added another reference in the introduction.